# SARS-CoV-2 specific T cell responses are lower in children and increase with age and time after infection

Carolyn A. Cohen[1], Athena P. Y. Li[1], Asmaa Hachim[1], David S. C. Hui [2], Mike Y. W. Kwan [3], Owen T. Y. Tsang[4], Susan S. Chiu[5], Wai Hung Chan [6], Yat Sun Yau [6], Niloufar Kavian[1], Fionn N. L. Ma[1], Eric H. Y. Lau [7], Samuel M. S. Cheng[8], Leo L. M. Poon [1,8], Malik Peiris [1,8] & Sophie A. Valkenburg [1✉]

SARS-CoV-2 infection of children leads to a mild illness and the immunological differences with adults are unclear. Here, we report SARS-CoV-2 specific T cell responses in infected adults and children and find that the acute and memory CD4[+] T cell responses to structural SARS-CoV-2 proteins increase with age, whereas CD8[+] T cell responses increase with time post-infection. Infected children have lower CD4[+] and CD8[+] T cell responses to SARS-CoV-2 structural and ORF1ab proteins when compared with infected adults, comparable T cell polyfunctionality and reduced CD4[+] T cell effector memory. Compared with adults, children have lower levels of antibodies to β-coronaviruses, indicating differing baseline immunity. Total T follicular helper responses are increased, whilst monocyte numbers are reduced, indicating rapid adaptive co-ordination of the T and B cell responses and differing levels of inflammation. Therefore, reduced prior β-coronavirus immunity and reduced T cell activation in children might drive milder COVID-19 pathogenesis.

[1] HKU-Pasteur Research Pole, School of Public Health, Li Ka Shing Faculty of Medicine, The University of Hong Kong, Hong Kong, SAR, China. [2] Department of Medicine and Therapeutics, Prince of Wales Hospital, Chinese University of Hong Kong, Hong Kong, SAR, China. [3] Department of Paediatric and Adolescent Medicine, Hong Kong Hospital Authority Infectious Disease Center, Princess Margaret Hospital, Hong Kong, SAR, China. [4] Infectious Diseases Centre, Princess Margaret Hospital, Hospital Authority of Hong Kong, Hong Kong SAR, China. [5] Department of Paediatric and Adolescent Medicine, The University of Hong Kong and Queen Mary Hospital, Hospital Authority of Hong Kong, Hong Kong SAR, China. [6] Department of Paediatrics, Queen Elizabeth Hospital, Hospital Authority of Hong Kong, Hong Kong SAR, China. [7] WHO Collaborating Centre for Infectious Disease Epidemiology and Control, School of Public Health, Li Ka Shing Faculty of Medicine, The University of Hong Kong, Hong Kong, SAR, China. [8] Division of Public Health Laboratory Sciences, School of Public Health, Li Ka Shing Faculty of Medicine, The University of Hong Kong, Hong Kong, SAR, China. ✉email: sophiev@hku.hk

A lack of pre-existing SARS-CoV-2-specific protective antibodies has led to the rapid global spread of the novel coronavirus, however the large majority of infections are reportedly asymptomatic or mild[1]. Previous COVID-19 infection may protect from reinfection[2,3] and neutralising antibodies are likely to play an important protective role[4]. However, the emergence of variant strains (e.g., 501Y.V2) suggests the possibility of escape from previous neutralising antibodies[5,6]. Antibody-based treatment of established infection has had a minimal beneficial effect on clinical outcome in COVID-19 patients[7] and may lead to the emergence of escape mutant variants[8]. Dysregulated innate immune responses, such as auto-interferon antibodies or delayed responsiveness have been reported in some severe COVID-19 cases but cannot account for the majority of severe infections[9–11]. Importantly, a coordinated cellular immune response has been key to the clinical resolution of SARS-CoV-2 infection[12].

Pre-existing cross-reactive antibodies elicited by exposure to endemic human common cold coronaviruses such as the related β-coronavirus OC43 and HKU-1, do not prevent infection with SARS-CoV-2[13,14]. Furthermore, pre-existing cross-reactive T cell immunity generated by common cold coronaviruses has also been detected in the majority of people[15], with epitope conservation mostly reported in the ORF1ab nonstructural proteins[16], but SARS-CoV-2 cross-reactive T cell responses have also been detected despite lower (< 67%) epitope homology[17]. Upon infection, T cell responses shift towards Spike and Nucleocapsid structural proteins[17,18]. However, cross-reactive CD4+ T cell responses have been reported as similar[17] or lower avidity and may be associated with worsening clinical outcomes[19]. In animal models of reinfection, Spike-specific CD8+ T cell responses can compensate for inadequate antibody responses and may provide an immune correlate of protection[20]. The magnitude of ORF1ab specific SARS-CoV-2 T cell responses during infection of adults does not differ with symptom severity but does associate with reduced duration of illness[18]. Therefore, determining the balance and specificity of SARS-CoV-2-specific T cell responses for structural, accessory, and nonstructural proteins may inform the COVID-19 response and pathogenesis.

Following mild COVID-19 infection SARS-CoV-2 specific memory B cells are established for at least 6 months with long-term stability that may be recruited upon reinfection[21]. T cells following SARS-CoV infection in 2003 have reassuringly been detected 17 years after infection[18]. Robust adaptive antibody and T cell responses have been reported in symptomatic and asymptomatic SARS-CoV-2 infected adults[22,23]. Although serum antibody response to the common cold coronaviruses may be long-lasting, reinfection is common one or more years after infection[14]. The severity of COVID-19 may be reduced by rapid and early recruitment of established immune responses[24–26]. The early and rapid recruitment of T follicular helper (Tfh) cells[27] drives early antibody development[24] by germinal B cell responses leading to increasing neutralising antibody titers, however increased disease severity is associated with higher viral loads and antibody titers[18]. The magnitude of the acute T cell responses in Middle Eastern Respiratory Syndrome (MERS), a related β-coronavirus, is negatively associated with the magnitude of the CD4+ T cell response and the duration of illness and thus antigen loads[28,29].

In a small family case study, children (n = 3) exposed to their SARS-CoV-2 infected parents displayed coordinated recruitment of total T cells and specific antibodies however infection was not able to be virologically confirmed[26]. Asymptomatic infection may represent a large proportion of SARS-CoV-2 infections, particularly in children. The immunological differences of cellular recruitment for children and adults have not been sufficiently characterised to determine the immunological basis of differing diseases severity and outcomes of COVID-19.

In Hong Kong, effective public health measures of a track, trace, quarantine of returned travellers, and testing of quarantined close contacts have led to the identification of RT-PCR confirmed asymptomatic infections, even in young children. In this study, we assessed the balance of specificity, memory phenotype, cytokine quality, and longitudinal stability of SARS-CoV-2 T cell responses in children (aged 2–13 years old) and adults with asymptomatic or symptomatic infection to address the role of T cells in the pathogenesis of milder disease in children.

## Results

**SARS-CoV-2-induced CD4+ T cell responses to structural proteins**. SARS-CoV-2 specific T cell responses were assessed from COVID-19 cases in children and adults, and in adult negative controls. SARS-CoV-2 consists of 4 structural proteins, an extensive ORF1ab which encodes 16 nonstructural proteins, and 7 accessory proteins. The relative expression of the structural proteins versus accessory and nonstructural proteins during SARS-CoV-2 virus replication may impact their immunogenicity. Cross-reactivity with common cold viruses[14] may also affect the magnitude of T cell responses elicited. Due to the limited cell numbers of our samples, peptide or protein-specific mapping was not possible. Therefore direct ex vivo CD4+ and CD8+ T cell responses were assessed for overlapping peptide pools of structural, accessory, and ORF1ab proteins respectively, (Fig. 1b) using IFNγ production, a key anti-viral cytokine as a read-out for specificity (Fig. 1c). Paired samples from SARS-CoV-2 infected adults at hospital admission (time 1) and discharge (time 2) showed an increase in structural specific IFNγ+ CD4+ T cells (Fig. 1d, p = 0.0012, Wilcoxon two-tailed matched-pairs test, Fig. 1f, fold change p = 0.0005, two-tailed one-sample Wilcoxon test) and to a lesser extent CD8+ T cells (Fig. 1e, p = 0.2579, Wilcoxon two-tailed matched-pairs test; Fig. 1f, fold change p = 0.0230, two-tailed one-sample Wilcoxon test).

To confirm the appropriate use of IFNγ production as a surrogate measure of virus-specific T cell responses, three assays were initially used. T cell responses from infected children and adults (memory time point samples, > day 14) and negative controls of both children and adults for CD4+ T cells were measured for IFNγ production, IL-4 production, and expression of activation-induced markers (AIM by CD40L+ CD69+ CD137+ OX40+, negative adults only) and CD8+ T cells responses were measured by IFNγ production and AIM expression (Fig. 1g). IFNγ and AIM assays also showed higher responses in infected adults compared with negative adults confirming assay specificity. All assays showed that infected adults had greater structural specific T cell responses than infected children (Mann–Whitney two-tailed test). However, the low magnitude of T cell responses directly ex vivo from infected children showed no significant difference to uninfected children. IFNγ CD8+ T cells responses further showed a difference between negative children and negative adults, but there was no difference between infected and control groups of either age group. Uninfected children had significantly lower IFNγ+ CD8+ T cell responses than uninfected adults (Fig. 1g), which is in agreement with our hypothesis that children have lower cross-reactive CD8+ T cell memory with lower prior common cold coronavirus exposure. IFNγ and IL4 T cell responses are distinct cell populations with no correlation between responses (Fig. 1h).

The magnitude of SARS-CoV-2 specific CD4+ (Fig. 2a) and CD8+ (Fig. 2b) T cells for structural, accessory, and ORF1ab proteins was compared between adult patients versus adult negative controls to establish assay specificity and cross-reactivity.

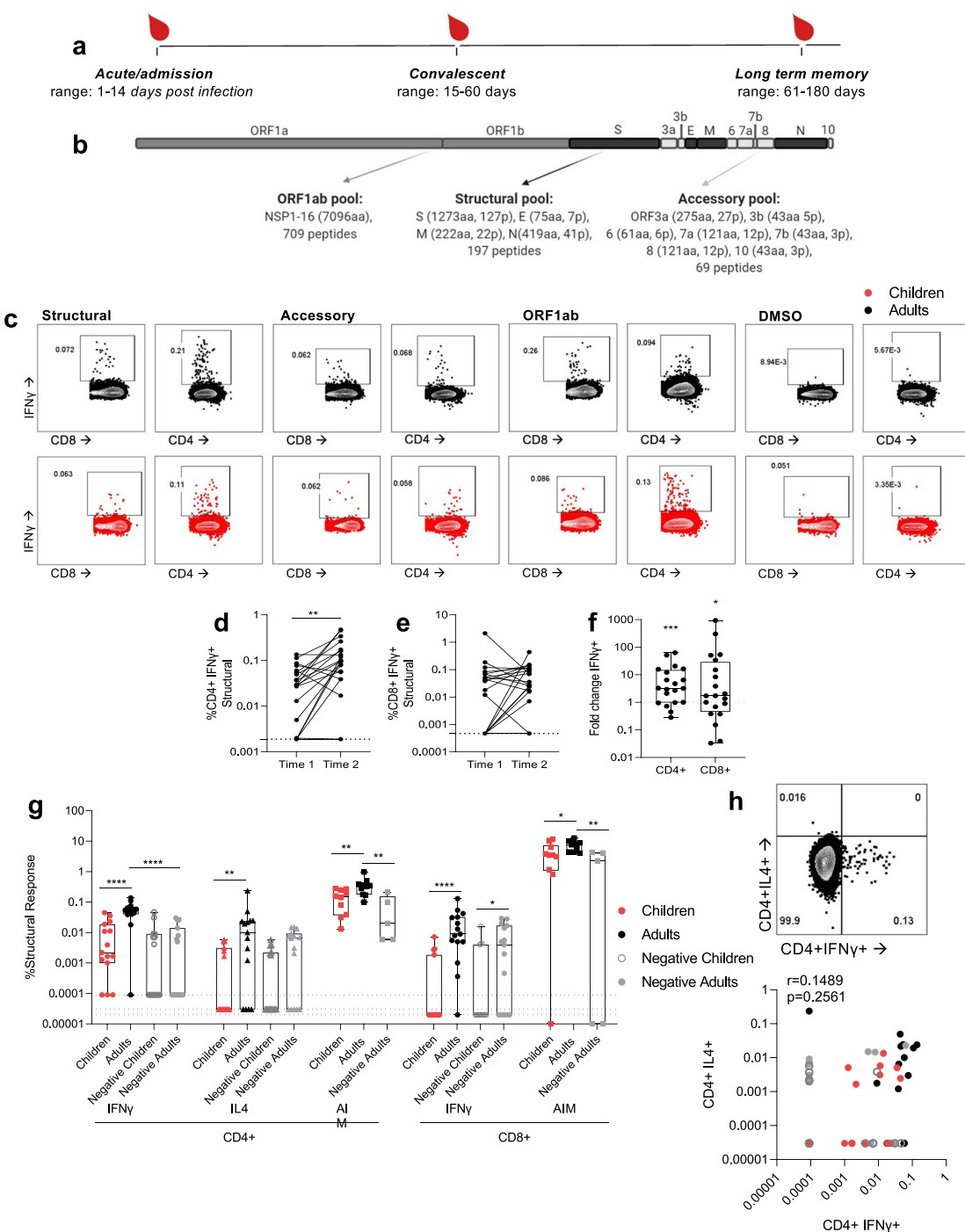

We then compared the T cell responses of the adult infections versus paediatric infections to define differences with age. The IFNγ$^+$ CD4$^+$ T cell responses towards structural proteins of SARS-CoV-2 were significantly increased in adults (mean±stdev: 0.0533 ± 0.0549%), compared to both children (0.0240 ± 0.0292%, $p = 0.0065$, Mann–Whitney two-tailed test) and adult negative controls (0.0013 ± 0.0005%, $p < 0.0001$, Mann–Whitney two-tailed test) (Fig. 2a). The majority of infected adults (94.3%) mounted structural-specific CD4$^+$ T cell responses (above DMSO background) (Fig. 2c), whilst only 79.4% of infected children and 50% of adult negative controls had such responses (Fig. 2c). Despite the higher magnitude of responses to structural proteins in infected adults than children, the proportion of responders against each peptide pool was not significantly different between

adults and children, except for structural CD8$^+$ T cell responses (Fig. 2c). Therefore, the majority of our later analyses focusses on structural specific T cell responses.

The accessory-specific CD4$^+$ T cell response was comparable in infected children, infected adults, and adult negative controls (Fig. 2a). In infected adults, the structural-protein-specific CD4$^+$ T cell responses (86.6%) contributed most to the SARS-CoV-2 specific response (Fig. 2d), than ORF1ab (9.6%) and accessory (3.8%) responses. By contrast, the SARS-CoV-2 specific response in infected children's CD4$^+$ T cell responses was dominated more by ORF1ab (51.8%) than structural specific responses (43.7%). Responses from adult negative controls that recognised SARS-CoV-2 peptides were predominantly specific for accessory peptides (90.1%), however the total response was

**Fig. 1 Infected children have lower CD4+ and CD8+ T cell responses than adults. a** Heparinised blood samples for PBMCs were collected from COVID-19 patients in Hong Kong during the course of infection and recovery. **b** Overlapping peptide pools of the whole SARS-CoV-2 proteome were generated to represent ORF1ab, structural, and accessory proteins with amino acids (aa) and peptides (p) per protein shown. **c** PBMCs from adults (black) and children (red) were stimulated with peptide pools or a DMSO control and IFNγ production of CD4+ and CD8+ T cells measured by flow cytometry (see Supplementary Figure 1 for gating strategy). Paired time points at hospital admission and discharge (time 1: mean 7.25 ± stdev 4.6 days post-infection, range 3–18; time 2: mean 13.4 ± stdev 4.4, range 6–21) for paired background (DMSO) subtracted structural specific IFNγ response of CD4+ (**d**) and CD8+ (**e**) T cells (n = 20 adults). A two-sided Wilcoxon test was used to determine differences **p < 0.01. Dotted lines represent the limit of detection following background subtraction (IFNγ of CD4+ = 0.0019, IFNγ of CD8+ = 0.00047). **f** The fold change of paired structural specific CD4+ and CD8+ T cells responses from (**d**, **e**), significance calculated using One-sample Wilcoxon test against a theoretical median of 1, *p < 0.05, **p < 0.01, ***p < 0.001. The dotted line at 1 indicates no fold change. The SARS-CoV-2 CD4+ (**g**) or CD8+ (**h**) T cell responses of COVID-19 children (n = 15), adults (n = 15) (mean± stdev:34 ± 11 days, range 14–57 days)) and negative children (n = 15) and negative adults (n = 15). Data are displayed as individual responses with box and whiskers plots representing the median, upper and lower quartiles, and minimum and maximum values against the structural peptide pool, measured by IFNγ production in CD4+ and CD8+ T cells, IL4 production in CD4+ T cells, and surface expression of the combination of CD40L, CD137, OX40, and CD69 activation-induced markers (AIM), with paired responses to DMSO subtracted. The dotted lines represent the lower limits of detection for ICS assays, determined as the smallest calculated value above the DMSO background response (IFNγ of CD4+ = 0.00009%, IL-4 of CD4+ = 0.00003%, IFNγ of CD8+ = 0.00002%). Comparisons between groups were performed using the Two-sided Mann–Whitney test, statistical differences are indicated by **p<0.01, ****p<0.0001. CD4+ T cells do not simultaneously produce IFNγ and IL-4 as shown by representative FACS plot and correlation (**h**). **d** **p = 0.0012, (**f**) ***p = 0.0005, *p = 0.0230, (**g**) ****p< 0.0001, < 0.0001, < 0.0001, **p = 0.0082, 0.0052, 0.0047, *p = 0.0243, 0.0355, **p = 0.0027.

very low in magnitude, at only 0.013±0.02% of CD4+ T cells (Fig. 2d).

Infected adults did not have significantly higher CD8+ T cell responses compared to adult negative controls (Fig. 2b) indicating cross-reactivity and little amplification of SARS-CoV-2 CD8+ T cell responses by infection (Figure 1e, f). But infected children had significantly reduced CD8+ T cell responses compared to infected adults for structural and ORF1ab responses (Fig. 2b).

Furthermore, the stratification of subjects for asymptomatic and symptomatic infection did not reveal any further significant differences for T cell response magnitude (Supplementary Figure 2a, b) or contribution of peptide specificities (Supplementary Figure 2c, d) between controls and COVID-19 adults and children.

However, baseline differences exist between adults and children for nonspecific T cell activation[30–32]. The baseline activation (by DMSO) (Fig. 3a) and overall maximum activation (by PMA/ ionomycin) (Fig. 3b) are lower in infected children. Overall background and maximum T cell responsiveness significantly increase with age in infected subjects (Fig. 3c, d). Adult negative controls had comparable background IFNγ induction compared to infected adults (Fig. 3a), but significantly higher maximum responsiveness (Fig. 3b). Whereas negative children had significantly higher nonspecific (for CD4+ T cells) and maximum activation than infected children, yet structural specific IFNγ+ T cell activation was not different between these groups (Fig. 1g). Therefore, normalisation of structural specific T cells by % of maximum PMA/ionomycin responses after background DMSO subtraction shows that infected adults continue to have higher virus-specific CD4+ and CD8+ IFNγ T cell responses than children, whilst negative adults also have a higher response than negative children. This normalised response shows a significant difference between infected and uninfected children, and infected and uninfected adults for the CD4+ but not CD8+ IFNγ T cell response (Fig. 3e). Overall, even with normalisation infected children still have lower CD4+ and CD8+ T cell responses than infected adults. The fine specificity of identifying low-frequency antigen-specific T cells directly ex vivo may be obscured through normalisation, and maximal activation is refractory to recent infection, therefore T cell responses should be considered directly ex vivo with paired DMSO background subtracted.

**Recruitment of early cellular responses**. Coordination of the early acute response to SARS-CoV-2 infection is important to drive innate responses[33], and early antibody production[24] for

improved patient outcomes. Therefore, we assessed the recruitment and activation of monocytes, total Tfh cells, and plasmablasts (also known as antibody-producing cells) during acute (< 14 days post-infection) and convalescent (15–57 days post-infection) SARS-CoV-2 infection in two separate experiments (Fig. 4). At acute time points, the total monocytes showed reduced responses in children compared to infected adults (Fig. 4b), furthermore children had reduced inflammatory type monocytes (Fig. 4c), where these have been found to also be elevated in COVID-19 patients, but lower in severe outcomes in adults[33,34]. Meanwhile, infected children and adults showed comparable levels of monocyte recruitment from bone marrow (CCR2) compared to infected adults (Fig. 4d). Overall, negative children and negative adults had equivalent monocyte populations (Fig. 4f–h).

The coordinated recruitment of circulating Tfh for germinal centre reactions and early antibody production by plasmablasts is associated with seroconversion[35]. The early activated (ICOS+ PD-1+) total Tfh response was significantly increased in infected children compared to adults and negative controls (Fig. 4e) and remained higher at convalescent time points (Fig. 4k), indicating sustained germinal centre reactions. Whilst plasmablast responses were increased in both children and adults compared to negative controls showing early B cell recruitment with infection (Fig. 4d). At convalescent time points, plasmablast responses in infected children were equivalent to uninfected children, but uninfected children also have significantly higher plasmablast responses than negative adults (Fig. 4j).

**T cell responses increase with time post-infection and age**. Longitudinal sample collection up to 180 days post-infection enabled us to determine the trend of T cell responses with time post-infection. Long-term stability of durable T cell immunity is likely important to minimise the symptom severity of reinfection with SARS-CoV-2. The CD4+ T cell response to structural peptides had stable responses post-infection (Fig. 5a) (r = 0.1475, p = 0.2265, Spearman two-tailed correlation), whilst structural-specific CD8+ T cell responses had a moderate significant trend for increased responses with time (Fig. 5B) (r = 0.4194, p = 0.0003, Spearman two-tailed correlation). This was also reflected in the acute fold changes of CD4+ and CD8+ T cell responses (Fig. 1f), which indicates the CD4+ T cell response is recruited early during SARS-CoV-2 infection (Fig. 1d), the CD8+ T cell response takes more time to build up with time post-infection. Furthermore, there was no difference in T cell

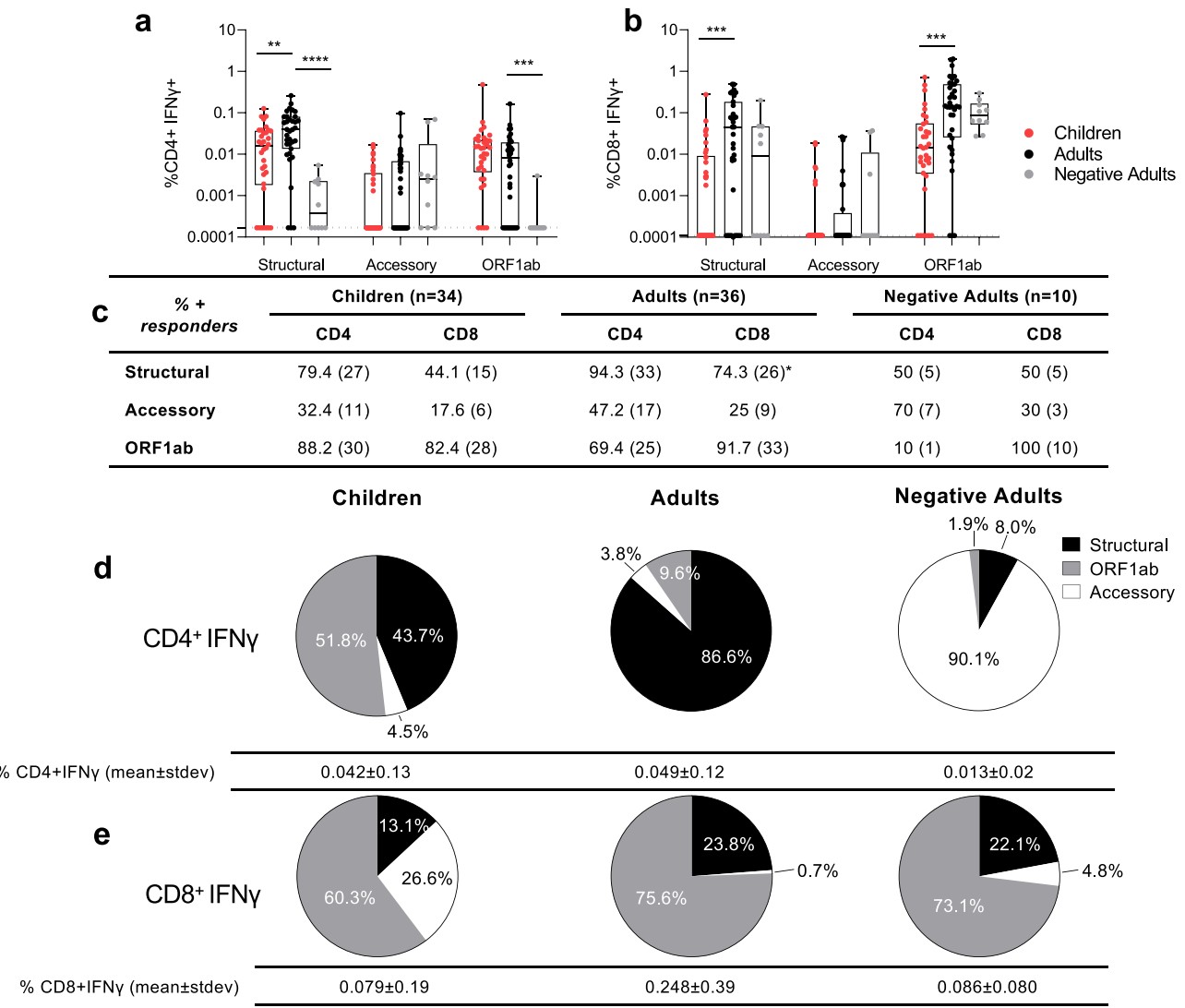

**Fig. 2 Specificity of T cell responses in adults and children.** The SARS-CoV-2 CD4+ (**a**) or CD8+ (**b**) T cell responses of COVID-19 children ($n = 34$), adults ($n = 36$) (mean±stdev: 42 ± 44, range 1–180 days) and negative adults ($n = 10$). Data are displayed as individual responses to each peptide pool with IFNγ production to paired DMSO subtracted, with box and whiskers plots displaying the median, upper and lower quartiles, minimum and maximum values. The dotted line represents the lower limit of detection, determined as the smallest calculated value above the DMSO background response (IFNγ of CD4+ = 0.00017%, IFNγ of CD8+ = 0.00011%). **a**, **b** Comparisons between groups were performed using two-sided Mann–Whitney test statistical differences are indicated by *$p<0.05$, **$p<0.01$, ***$p<0.001$. Values above the limit are used to classify participants as responders and presented as a percentage with the numbers of responders in brackets (**c**). Differences between children ($n = 34$) and adults ($n = 36$) from all time points (1 to 180 days post symptom onset) were determined by Fisher's exact test and displayed in the adults column where *$p < 0.05$. Pie charts show the proportion of total IFNγ+ CD4+ (**d**) and CD8+ (**e**) SARS-CoV-2 responses with DMSO subtracted in children ($n = 34$), adults ($n = 36$) and negative adults ($n = 10$) (from **a**, **b**). Values below the limit of detection assigned the value of 0. (**a**) **$p = 0.0065$, ****$p < 0.0001$, ***$p = 0.0008$, (**b**) ***$p = 0.0003$, 0.0001.

exhaustion (by PD-1 expression) between infected adults and children at either acute or memory responses (Fig. 4).

The fold change of response magnitude for paired acute responses (< 14 days) to memory time-points (> 14 days) (Fig. 5c, d), showed comparable fold changes in children and adults for CD4+ or CD8+ T cell response to most viral proteins. Only accessory-specific CD8+ T cell responses had a significant decrease in infected children (Fig. 5d). Whilst the acute structural specific CD4+ T cell response was significantly increased in adults compared to negative controls (Fig. 5e), the memory CD4+ and CD8+ T cell response was significantly lower in children compared to infected adults (Fig. 5e, h), resulting in a trend for significantly increased T cell responses with age (Figure 5f, I, j), excluding acute CD8+ T cell responses (Figure 5g).

The difference in magnitude of T cell responses with age and time indicates functional differences in T cell recruitment and differentiation, therefore we assessed cytokine polyfunctionality and memory phenotypes. Cytokine polyfunctionality is associated with increased protection from infection for multiple viruses[36,37], and associated with cellular division and terminal differentiation[38]. Whilst differentiation of T cell memory phenotypes occurs early during infection and can reflect the extent of inflammation[39], impacting recall capacity long-term[40] to infected tissues[41].

Cytokine polyfunctionality of structure-specific T cells (Figure 6a, b) was comparable between adults and children at acute (< 14 days), convalescent (15–60 days), or memory (61–180 days) time points (Fig. 6c), therefore on per-cell basis adults and children had comparable cytokine responses. The phenotype of

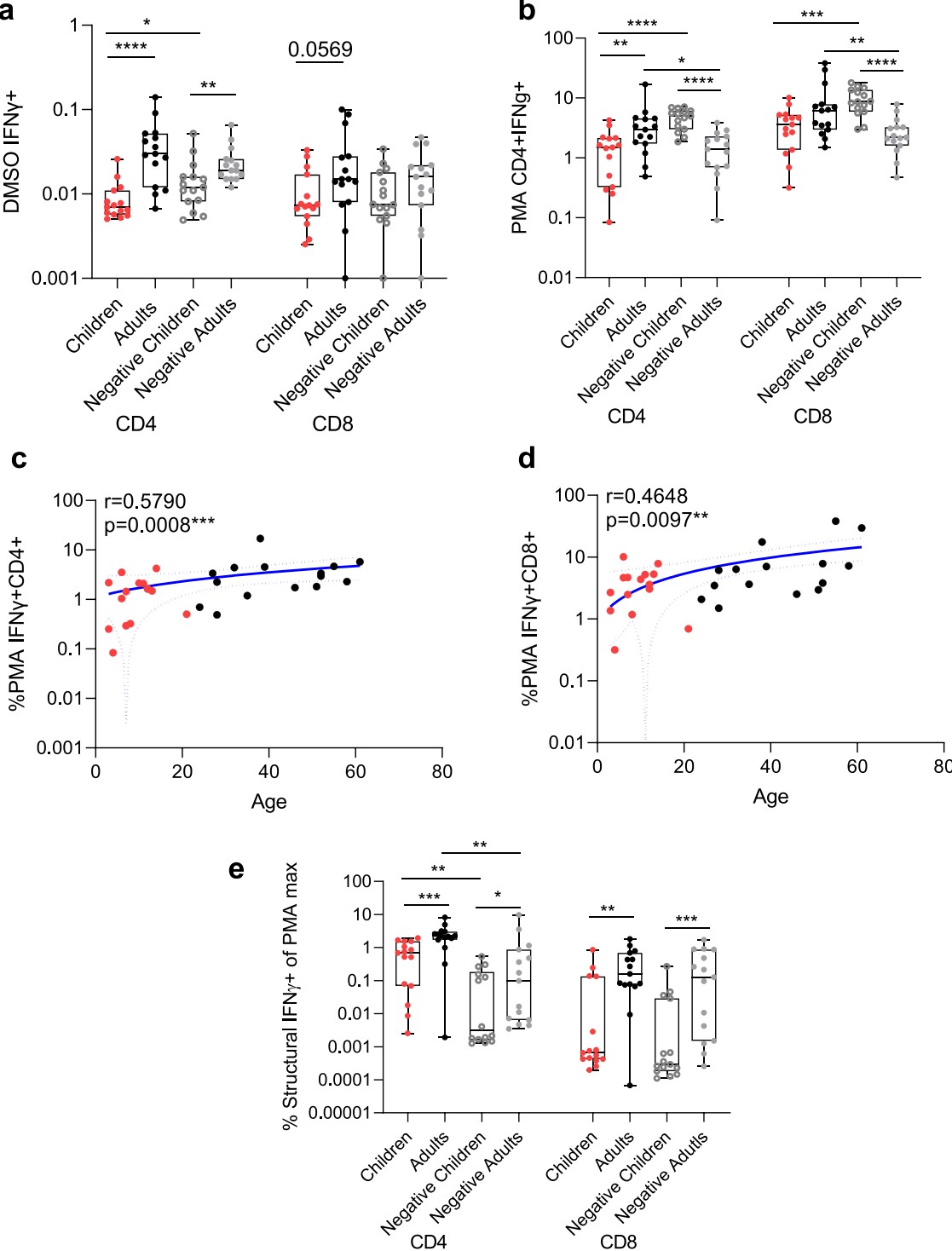

**Fig. 3 Non-specific T cell responses increase with age in infected donors.** CD4+ and CD8+ T cell responses for (**a**) background (by DMSO stimulation) and (**b**) maximum (by PMA/Ionomycin stimulation) in children ($n = 15$), adults ($n = 15$) from convalescent/ memory time points (mean ± stdev 34 ± 11, range: 14–57 days post symptom onset), and uninfected negative children ($n = 15$) and adult ($n = 15$) controls. Comparisons were made by Mann–Whitney test where **$p < 0.01$, ***$p < 0.001$, ****$p < 0.0001$. Correlation of age with CD4+ (**c**) and CD8+ (**d**) T cell responses by PMA/ionomycin stimulation. Two-sided Spearman's test was used to calculate r values, and statistical significance is displayed as ***$p < 0.001$. **c, d** Blue lines of linear regression represent the overall trend with dotted lines showing 95% confidence intervals. Black dotted lines represent the limit of detection (IFNγ of CD4+ = 0.00009% IFNγ of CD8+ = 0.00003%). **e** The structural peptide pool response for CD4+ and CD8+ T cells in adults and children (from Fig. 1g) normalised to a paired maximum IFNγ production from (**b**) PMA/ionomycin stimulation. Comparisons similarly made by Two-sided Mann–Whitney test where *$p < 0.05$, **$p < 0.01$, ***$p < 0001$. (**a, b, e**) Data is representative of individual values with box and whiskers plots showing the median, upper and lower quartiles, and minimum and maximum. (**a**) *$p = 0.0463$, **$p = 0.0054$, ****$p < 0.0001$, (b) CD4 *$p = 0.0164$, **$p = 0.0086$, ****$p < 0.0001$, CD8 **$p = 0.0057$, ***$p = 0.0002$, ****$p < 0.0001$, (**e**) CD4 *$p = 0.0259$, **$p = 0.0011$, 0.0049, ***$p = 0.0005$, CD8 **$p = 0.0049$, ***$p = 0.0008$.

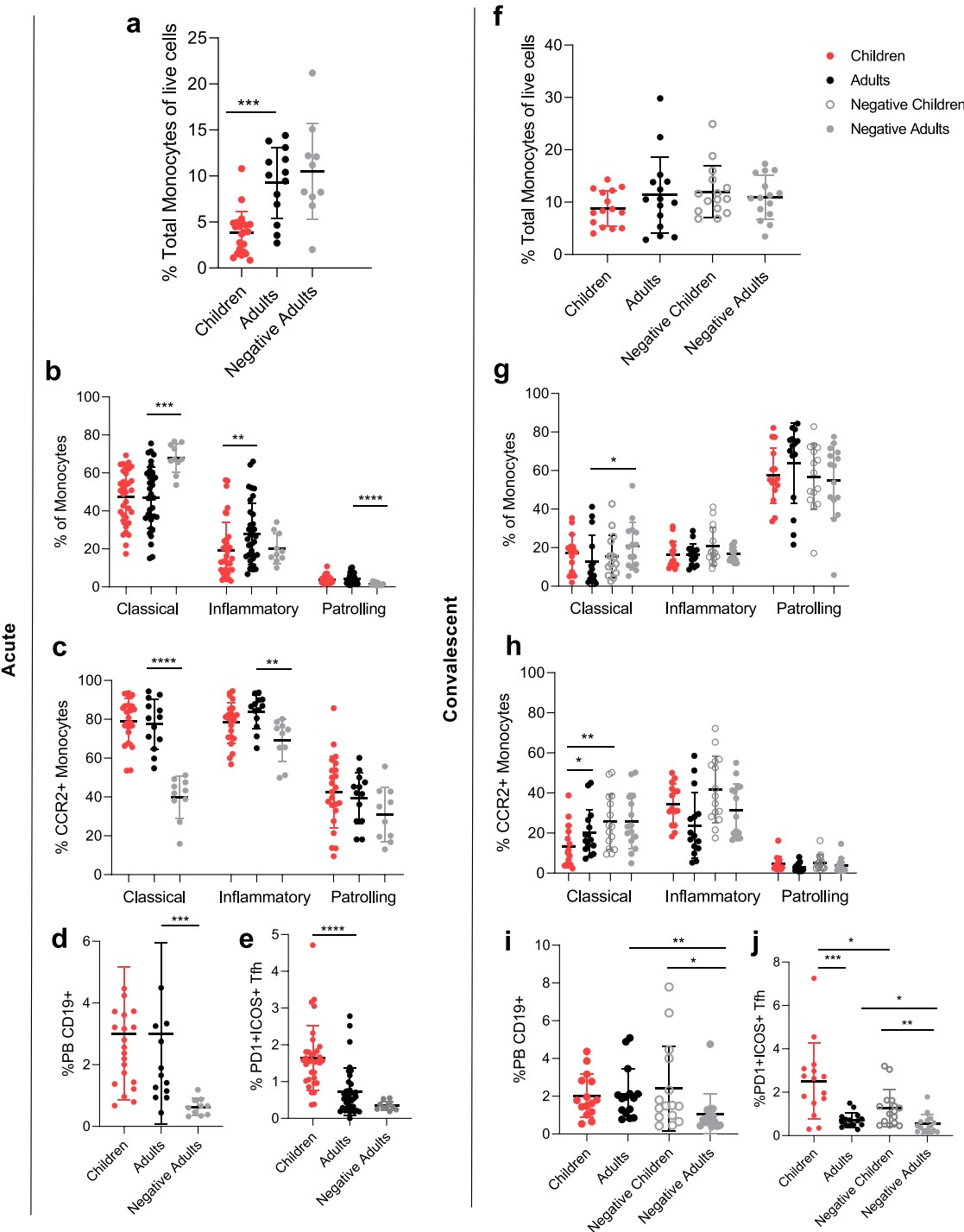

**Fig. 4 Cellular recruitment of Tfh cell, plasmablasts, and monocytes.** Early (< day 14) recruitment of innate and adaptive cells was measured by flow cytometry (see Supplementary Figure 3 for gating strategy) for COVID-19 children ($n = 22$), adults ($n = 13$), and negative controls ($n = 10$) (**a–e**). Convalescent/ memory samples of children ($n = 15$), adults ($n = 15$), and negative adults ($n = 15$) were also tested alongside negative children ($n = 15$) (**f–j**). Total monocytes (**a**, **f**), monocyte phenotype (**b**, **g**), and activation of monocytes (**c**, **h**). Total plasmablast (**d**, **i**) and activated T follicular helper cell (**e**, **f**). Data represents the individual response, mean ± SD. Statistical differences were determined using a two-sided Mann–Whitney test between children and adults, adults and negative adults, and children and negative children where **$p < 0.01$, ***$p < 0.001$, ****$p < 0.0001$. **a** ***$p = 0.0002$, (**b**) **$p = 0.0054$, ***$p = 0.0001$, ****$p < 0.0001$, (**c**) **$p = 0.0020$, ****$p < 0.0001$, (**d**) ***$p = 0.0001$, (**e**) ****$p < 0.0001$, (**g**) *$p = 0.0321$, (**h**) *$p = 0.0203$, **$p = 0.0086$, (**i**) *$p = 0.0124$, **$p = 0.0016$, (**j**) *$p = 0.0489, 0.0235$, **$p = 0.0013$, ***$p = 0.0004$.

structure-specific T cells at memory time points (Fig. 6d), however showed that children had reduced T effector memory (TEM) CD4$^+$ T cells compared to infected adults (Fig. 6e). The phenotype of structure-specific CD8$^+$ T cells was comparable (Fig. 6f).

**Prior common cold coronavirus immunity and cellular responses**. The level of coronavirus Spike-specific IgG was determined at early time points (< 14 days) of SARS-CoV-2 infection, to determine if pre-existing immunity impacted T cell responses. The magnitude of α-coronavirus 229E and NL63-

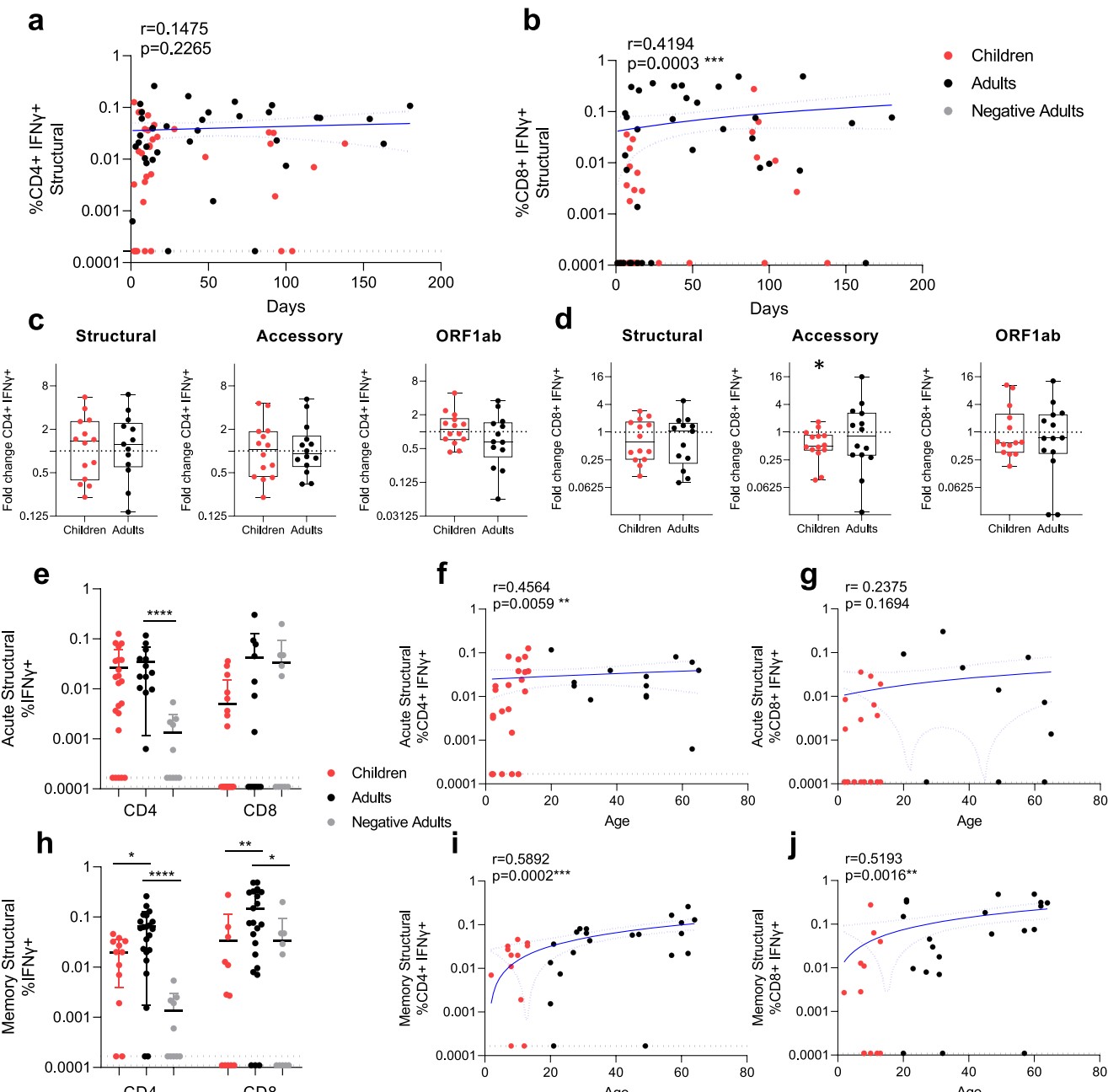

**Fig. 5 SARS-CoV-2 specific T cell responses increase over time and age.** Correlation of IFNγ responses for CD4+ (**a**) and CD8+ (**b**) T cells against the structural peptide pool with children (red) (n = 34) and adults (black) (n = 36) (with background IFNγ production to DMSO subtracted), against days post symptom onset. Black dotted lines represent the limit of detection (IFNγ of CD4+ = 0.000167 (**a**), IFNγ of CD8+ = 0.00011(**b**)). Fold change of IFNγ CD4+ (**c**) and CD8+ (**d**) T cell responses were calculated as the later time point (mean ± stdev: 32.8 ± 35.7 days, range: 9–138) over admission time point responses (mean±stdev: 7.6 ± 4.2, range: 2–15)) in response to the structural, accessory and ORF1ab peptide pools in children and adults from two independent experiments (children n = 14, adults n = 14). Data is representative of individual data points with boxes and whiskers graphs showing the median, upper and lower quartiles, minimum and maximum. One-sample Wilcoxon tests were used for determining significance of fold changes, where *p<0.05. Acute (samples < 14 days post symptom onset, mean ± stdev: 8.0 ± 3.8, range: 1–14, n = 22 children, n = 14 adults) (**e–g**), and convalescent/memory (**h–j**) (mean ± stdev: 70.5 ± 41.9, range: 15–180 days post symptom onset, n = 12 children, n = 22 adults) IFNγ structural specific (**f, i**) CD4+ and (**g, j**) CD8+ T cell responses and negative controls (n=10). Data in e and h show individual data points with mean±SD. For statistical comparisons between children and adults, or adults and negatives, two-tailed Mann–Whitney tests were performed, *p < 0.05, **p < 0.01, ***p < 0.001, ****p < 0.0001. The magnitude of the acute (from e) and memory (from h) structural IFNγ CD4+ (**f, i**) and CD8+ (**g, j**) T cell response with age. (**a, b, f, g, i,** and **j**) r and p values are calculated using two-tailed Spearman's correlation and *p < 0.05, **p<0.01, ***p<0.001, ****p<0.0001. Blue lines of linear regression represent the overall trend, and blue dotted lines show the upper and lower 95% confidence intervals. All data points are individual responses minus paired background IFNγ response to a DMSO control. (**d**) *p = 0.0245, (**e**) ****p < 0.0001, (**h**) *p = 0.0162, 0.0219, **p = 0.0074, ****p < 0.0001.

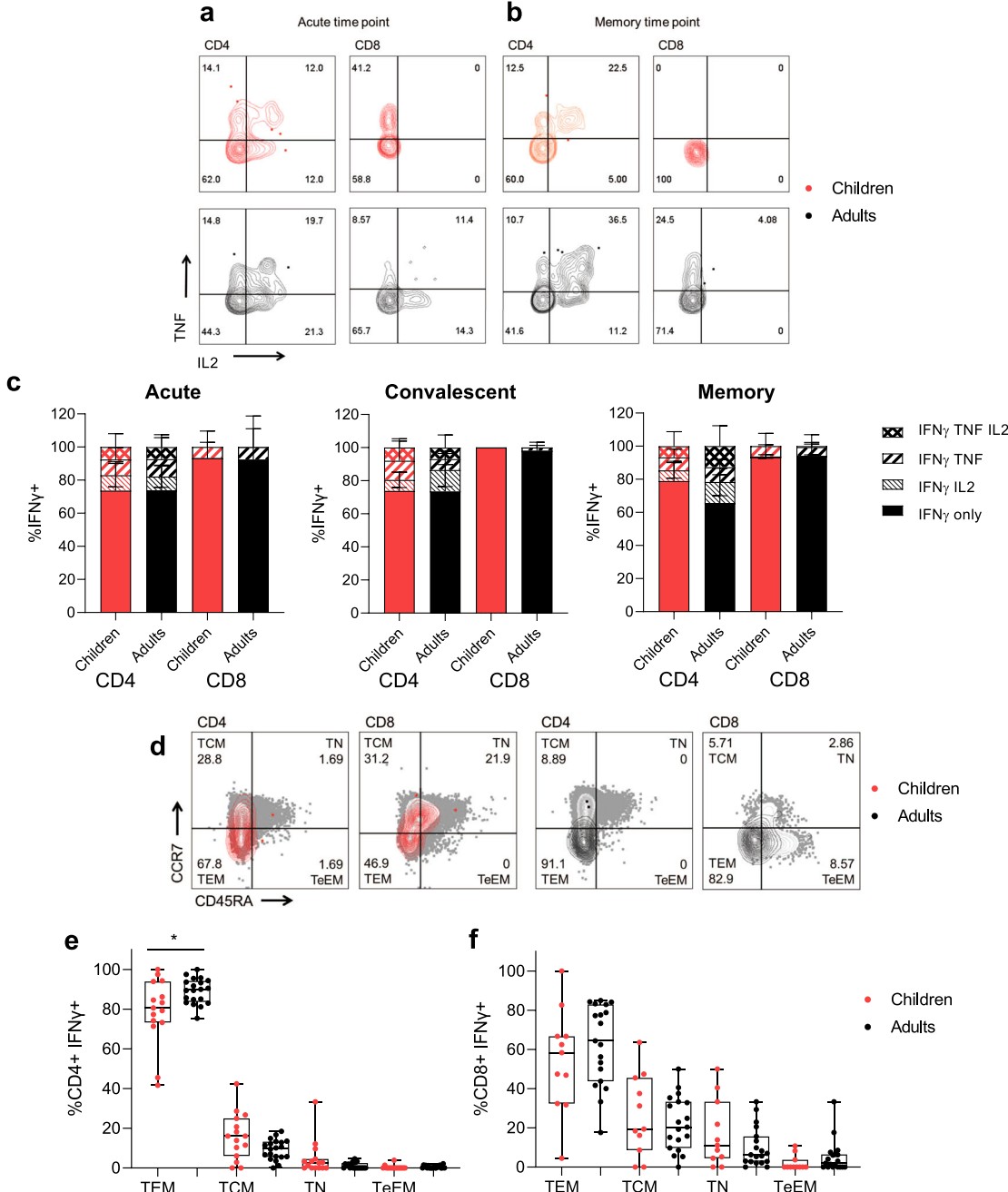

**Fig. 6 Cytokine polyfunctional quality and memory phenotype.** Representative FACS plots of TNF and IL2 producing IFNγ $^+$ CD4$^+$ and CD8$^+$ T cells of children (red) and adults (black) at acute (d < 14) (**a**) and memory (child: 118 days, adult: 94 days) (**b**) time points. (**c**) The proportion of IFNγ producing CD4$^+$ and CD8$^+$ T cells which are single, double, or triple cytokine producers at acute (< 14 days), convalescent (15–60 days), or memory (61–180 days) time points post symptom onset. Bars represent the mean values in infected children and adults, while error bars represent SD. Kruskal–Wallis test for multiple comparisons was carried out to compare each group between children and adults. **d** Representative FACS plots showing memory phenotypes of IFNγ $^+$ CD4$^+$ and CD8$^+$ T cells based on the expression of CCR7 and CD45RA. Sections are T effector memory (TEM), central memory (TCM), terminal effector memory (TeEM), or naïve (TN). Memory phenotype responses in IFNγ $^+$ CD4$^+$ (**e**) and CD8$^+$ (**f**) T cells of responders at later time points (15–180 days post symptom onset). Data shows individual values, box and whiskers plots median, upper and lower quartiles, minimum and maximum values. Comparisons between children (n = 15) and adults (n = 20) in each group were performed using the Mann–Whitney test, (**e**) *p = 0.0400.

specific IgG was comparable between infected children and adults and adult negative controls (Fig. 7a), whilst β-coronavirus HKU-1 and OC43-specific IgG was significantly lower in infected children than infected adults (Fig. 7b). Furthermore, there was no difference in OC43-IgG responses between symptomatic or asymptomatic infections, with significance only being seen between symptomatic adults and children (Fig. 7c). There was a

significant correlation of OC43-IgG responses with age (Fig. 7d) (r = 0.6466, p = 0.0002). However, there was no direct significant correlation between OC43-IgG responses and structure-specific CD4$^+$ (p = 0.1027, Fig. 7e) or CD8$^+$ T cells (p = 0.9729, Fig. 7f). However, there was a borderline moderate negative correlation between OC43-IgG and early acute activated Tfh responses (Fig. 7h) (r = −0.3326, p = 0.0779)[15].

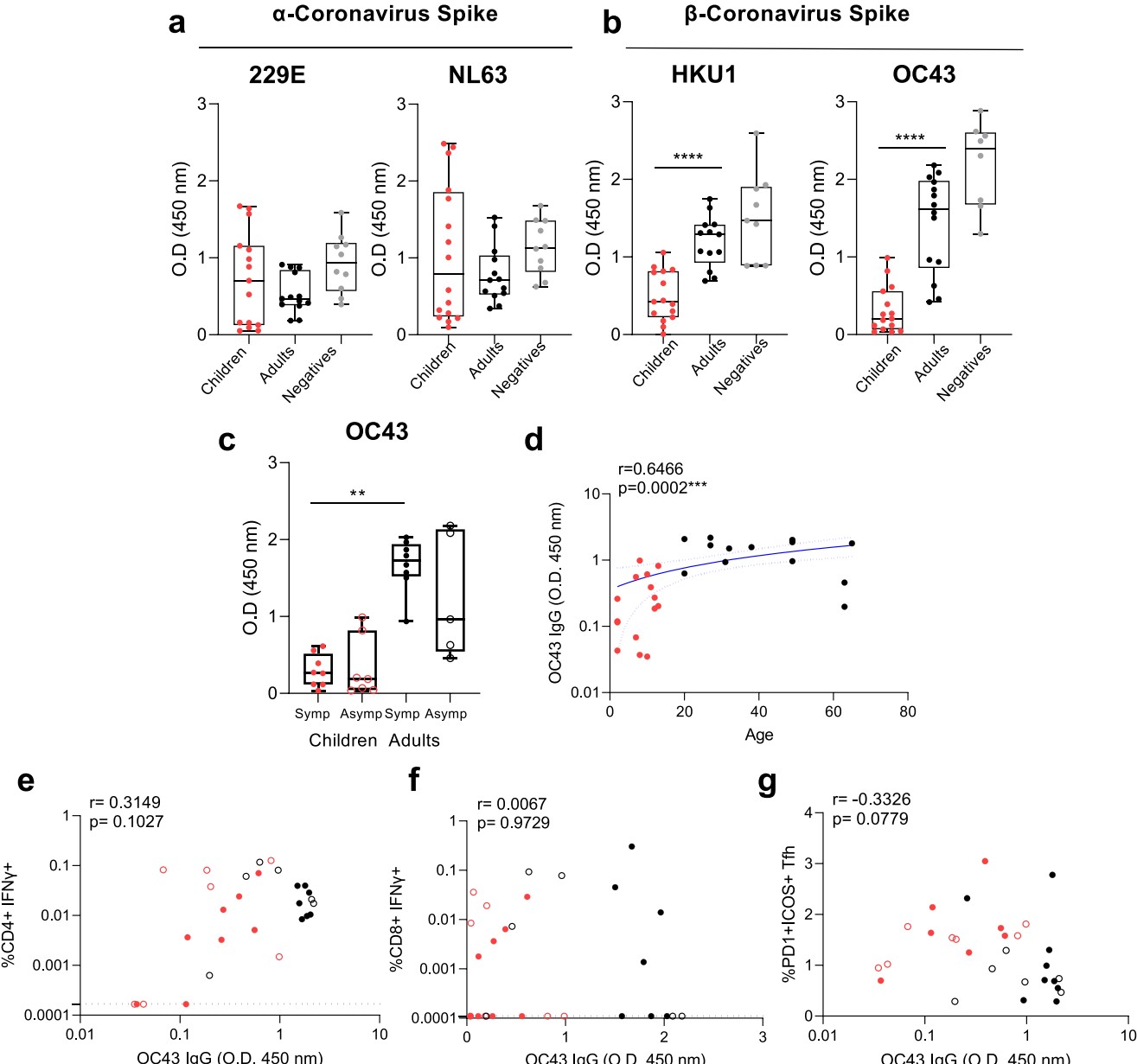

**Fig. 7 Previous exposure to common cold β-coronaviruses and T cell responses.** Total IgG responses to the Spike protein (S1 + S2) of common cold α (229E, NL63) (**a**) and β (HKU1, OC43) (**b**) coronaviruses measured by ELISA from acute time points (mean ± stdev: 8 ± 3.8, range: 2–14 days post-infection). **c** Stratification of OC43 IgG response by symptomatic (closed circles, $n = 8$ children, $n = 8$ adults) and asymptomatic (open circles, $n = 8$ children, $n = 5$ adults). **a–c** Data is representative of individual donor responses with background subtracted (nonspecific protein block), and displayed with box and whiskers plots of the median, upper and lower quartiles, and minimum and maximum values. Comparison between children ($n = 15$) and adults ($n = 14$) or adults negative controls ($n = 10$) was performed using two-tailed Mann–Whitney test where **$p < 0.01$, ***$p < 0.001$, ****$p < 0.0001$. **c** Multiple comparisons between symptomatic and asymptomatic adults and children were carried out using Kruskal–Wallis tests, where **$p < 0.01$. **d** Correlation of OC43 IgG and age. A blue line of linear regression represents the overall trend, and blue dotted lines show the upper and lower 95% confidence intervals. Correlation of structural SARS-CoV-2 specific IFNγ+ CD4+ (**e**) or CD8+ (**f**) T cell responses and OC43 Spike IgG. Correlation of activated Tfh and OC43 (**g**) Spike IgG. R values are determined using Spearman's correlation and statistically significant correlations are displayed as ***$p < 0.001$. Dotted lines indicate the limit of detection following subtraction of DMSO from T cell response. **b** ****$p < 0.0001$, < 0.0001, (**c**) **$p = 0.0062$.

## Discussion

SARS-CoV-2 infection of children is associated with milder clinical outcomes than adults, and the immune mechanisms are unknown. Several hypotheses have been proposed to explain these differences such as innate cell recruitment and impairment by autoantibodies[11], mobilisation of antibody responses, differing levels of pre-existing cross-reactive immunity by common cold coronavirus exposure[13], or baseline total IgM levels[42]. However, the SARS-CoV-2 T cell compartment in children has so far been under studied[26]. Viral loads[43] and neutralising antibody titers[44,45] are reportedly comparable when age is accounted for, however data is more limited in children. Viral loads, neutralising antibody titers[46], and T cell responses[47] impact clinical severity of COVID-19.

Cross-reactive T cell responses in unexposed adults have been mapped to have been NSPs of ORF1ab and Spike[16], whilst recent infection boosts structural Spike and N specific T cells[17,18]. The specificity of SARS-CoV-2 antibody landscapes differs in infected

children to adults[44,48], with an increased contribution by accessory proteins and lower total magnitude of responses[48], whilst the ORF1ab antibody response is under characterised. SARS-CoV-2 antibody landscapes indicate that the specificity and balance of the adaptive immune responses in children are different from adults.

Overall, we found total IFNγ CD4+ and CD8+ T cell responses are significantly lower in SARS-CoV-2 infected children than adults against the viral structural proteins, and in CD8+ T cells against ORF1ab proteins. Whilst infected adults had markedly higher structural specific CD4+ T cell responses than negative adults, it was only after normalisation that infected children's responses showed a significant increase compared to uninfected children, however these responses were still lower than infected adults. However, the T cell responses from infected children (at memory timepoints) had lower maximal activation compared to uninfected children negative controls. This may indicate T cell activation is refractory based on recent infection[49], and that SARS-CoV-2 infected children have dampened T cell responses.

The differences between infected children and adults may be due to differences in prior immunity to seasonal human coronaviruses through infection[50], resulting in qualitative differences in antigen-experienced CD4+ T cell responses in children. Children experience greater fold changes in influenza-specific T cell responses compared to adults during live attenuated influenza vaccination[51], whereas we found lower SARS-CoV-2 T cell response magnitudes in children, however their fold changes and polyfunctional cytokines of the T cell responses was comparable between adults and children. Therefore there is equal recruitment of SARS-CoV-2 T cell responses in adults and children but likely different baseline levels of cross-reactive responses to recruit from, which is also indicated by increased Tfh recruitment in children for driving antibody responses, higher effector memory T cells in adults, and higher β-coronavirus OC43 specific IgG in adults. The smaller magnitude of SASR-CoV-2 memory T cell responses in children than adults, may imply a weaker long-term memory response in children potentially impacting outcomes at reinfection. Indeed, we found significantly lower levels of β-coronavirus specific antibodies in infected children than adults, and there was a significant trend for both increased SARS-CoV-2 specific T cell responses and OC43-specific IgG with increasing age. Recently, similar results were found in healthy adults as HKU-1 IgG showed an increasing trend with SARS-CoV-2 specific T cell responses of memory phenotype in uninfected adults[15]. A borderline trend for decreasing acute activated Tfh with higher OC43-specific IgG levels also suggests a greater importance CD4+ T cell recruitment in more immunologically naïve settings, and as β-coronavirus specific IgG levels increase there is a decreasing drive for Tfh recruitment. Only the quantification of baseline T cell responses specific for common cold viruses and subsequent exposure to SARS-CoV-2 in further studies, such as in human cohort transmission settings or animal models, will determine if prior β-coronavirus immunity, based on T or B cells, has a protective role in COVID-19.

The quality of T cell responses, assessed by SARS-CoV-2 specific T cell polyfunctional cytokine production and exhaustion marker (PD-1) expression, was equivalent between children and adults, reflecting comparable division and terminal differentiation. The matched quality of response but the higher threshold for IFNγ production by T cells in children may drive a less inflammatory environment that promotes more mild COVID-19 outcomes in children. There was different recruitment of innate and adaptive cellular responses in adults and children during SARS-CoV-2 infection likely driven by a difference in inflammatory milieu despite comparable symptom severity (mild/asymptomatic in our study). Children had increased Tfh recruitment, comparable plasmablast responses, but reduced inflammatory monocytes, specific CD4+ and CD8+ T cell responses, in both magnitude and proportion of responders. Further mechanistic studies are needed to define the basis of immunological differences between T cell responses of children and adults indicated in our study.

We found that children had increased activated Tfh responses compared with both adults and negative children, but lower IFNγ+ CD4+ T cells than adults. Therefore the CD4+ T cell compartment is modulated by SARS-CoV-2 infection, and more so than the CD8+ T cell response which is lagging behind and is only boosted at convalescence. In the early stages of infection, infected adults had significantly higher CD4+ T cell responses to viral structural/ORF1ab proteins but CD8+ T cell responses were unremarkable. CD8+ T cell responses increased later in the course of the infection, suggesting that they may not be playing a major role in the recovery from the acute illness. This may also explain why patients continue to shed virus RNA detectable by RT-PCR for a prolonged period of time during convalescence[52]. The persistence of virus replication may also drive the SARS-CoV-2 CD4+ T cell response to also remain stable with time.

The contribution of different virion structural and nonstructural proteins reflects MHC processing access during viral replication, whereby MHCII access to structural proteins elicited substantial CD4+ T cell responses in adults, whereas in children the CD4+ T cell response was predominantly ORF1ab specific. The imbalance of peptide specificities for nonstructural proteins for children's CD4+ T cell compartment may indicate either different virus replication and pathogenesis at the cellular level or incomplete recruitment of de novo CD4+ T cell responses. Previously, in MERS-CoV infection, the magnitude of the CD4+ T cell response is proportional to virus replication and duration of illness[29]. This is consistent with the mild outcomes of COVID-19 in children and reduced T cell responses reported here in our study of mild and asymptomatic SARS-CoV-2 infection. We cannot attribute the differences in T cell response magnitude with the severity of illness in children to adults, unlike other reports[47], as the majority of both infections we studied are mild or asymptomatic. Therefore children have reduced SARS-CoV-2 T cell responses due to lower baseline immune activation, and further research is still needed to discern the protective role of T cells in COVID-19.

## Methods

**Study population and clinical samples**. Our study used samples from 24 children and 45 adults with RT-PCR confirmed SARS-CoV-2 infection in Hong Kong (Table 1). The days after onset of symptoms (for symptomatic infections) and days after first RT-PCR confirmation (for asymptomatic infections) were noted. All symptomatic or asymptomatic RT-PCR confirmed infections were hospitalised. Heparinised blood was collected at hospital admission (range: 1–14 days post symptom onset and/or RT-PCR confirmed infection), at discharge (range: 6–60 days), and at regular intervals after discharge for convalescent and long-term memory (range: 61–180 days) (Fig. 1a). We used samples from a total of 45 adults (mean±stdev: 43.1 ± 13.7, range: 20–65 years) and 24 children (8.1 ± 3.9, 1.92 (23 months)−13 years). We had 95 longitudinal samples from 46 subjects with 2 to 3 sampling time-points and 55 early acute time-points samples (< day 14) (Fig. 1a). Samples of comparable time-points were used from children (32.5 ± 40.4, 2–138 days) and adults (28.9 ± 39.6, 1–180 days) (Table 1). In experiments that include uninfected children controls, we used blood samples from a further 15 SARS-CoV-2 infected children (7.4 ± 4.4 years, 7 months–14 years) and 15 adults (41.7 ± 12.5, 24–61 years).

The study was approved by the institutional review board of the respective hospitals, viz. Kowloon West Cluster (KW/EX-20-039 (144-27)), Kowloon Central/ Kowloon East cluster (KC/KE-20-0154/ER2) and HKU/HA Hong Kong West Cluster (UW 20-273, UW20-169), Joint Chinese University of Hong Kong-New Territories East Cluster Clinical Research Ethics Committee (CREC 2020.229). All patients, children, and their parents provided informed consent. The collection of SARS-CoV-2 seronegative adult negative control blood donors (37.6 ± 13.0, 19–57 years) was approved by the Institutional Review Board of The Hong Kong University and the Hong Kong Island West Cluster of Hospitals (UW16-254).

**Table 1 Summary of cohort information.**

| SARS-CoV-2 RT-PCR+ | | | | | Negative controls | |
|---|---|---|---|---|---|---|
| | n (%) | Children | Adults | P value | Children | Adults |
| | Total donors | 24 | 45 | | 15 | 10 |
| | Age (mean ± stdev, range) | 7.8 ± 3.9, 1.92–13 years | 43 ± 4.0, 20–65 years | <0.0001 | 10.3 ± 3.2, 2–14 years | 37.6 ± 13.0, 19–57 years |
| | Female (%) | 54% (13) | 52% (23) | >0.9999 | 67% (10) | 40% (4) |
| Symptom Severity | Asymptomatic | 38% (9) | 20% (9) | | N/A | N/A |
| | Mild/ Moderate | 62% (15) | 80% (36) | 0.1523 | N/A | N/A |
| | Severe/ Critical | 0% | 0% | | N/A | N/A |
| *Sample time point information – days post symptom onset (n=, mean ± stdev, range)* | | | | | | |
| | All time points | n = 44 36 ± 38, 2–138 days | n = 75 29 ± 40, 1–180 days | 0.262 | N/A | N/A |
| | Acute time points | n = 22 8 ± 3.8, 2–14 days | n = 44 8 ± 4.0, 1–14 days | 0.949 | N/A | N/A |
| | Convalescent time points | n = 12 35 ± 10.9, 15–48 days | n = 19 26 ± 12.7, 15–53 days | 0.074 | N/A | N/A |
| | Long-term memory time points (d > 60) | n = 8 103 ± 38.7, 89–138 days | n = 12 111 ± 35.8, 67–180 days | 0.926 | N/A | N/A |

NB: *P* values are calculated to compare adults and children using Fisher's exact test to compare sex and symptom severity, and using Mann–Whitney to compare sample timepoint information. Samples from SARS-CoV-2 infected children and adults, and negative controls forming a cohort where samples were used in multiple cellular and ELISA assays.

SARS-CoV-2 seronegative children's control blood donors (*n* = 15) were recruited from immunocompetent children from renal, endocrine, and blood clinics (10.3 ± 3.2, 2–14 years) who were donating blood for non-infection related purposes. Informed consent was given by patients and parents and the collection of these samples was approved by HKU/HA Hong Kong West Cluster Hospitals (UW 20-273, UW20-169).

Plasma was isolated, stored at −80 ℃, and heat-inactivated (HI) at 56 ℃ for 30 min upon testing. Peripheral blood mononuclear cells (PBMC) were isolated by Ficoll-Paque (GE Healthcare) separation using Leucosep tubes (Greiner Bio-one) and cryopreserved in liquid nitrogen for batched analysis.

**SARS-CoV-2 overlapping peptide pools for T cell stimulation**. An overlapping peptide library was made covering the whole SARS-CoV-2 proteome with 20 amino acid (aa) length and 10 aa overlap (Genscript). The amino acid sequence of the peptide pools was based on βCoV/Hong Kong/VM20001061/2020 strain (GISAID ID: EPI_ISL_412028). Peptides were dissolved in deionised water, 10% acetic acid, or DMSO according to their biochemical properties and charge. A pool of 197 peptides representing Structural proteins: from S (1273aa, 127 peptides), N (419aa, 41 peptides), E (75aa, 7 peptides), M (222aa, 22 peptides), with a DMSO concentration of 0.6%. The ORF1ab peptide pool consisted of 709 peptides for the NSP1-16 proteins (7096aa), with a DMSO concentration of 2.1%. An accessory peptide pool of 69 peptides for the ORF3a (275aa, 27 peptides), ORF3b (43aa, 5 peptides), ORF6 (61aa, 6 peptides), ORF7a (121aa, 12 peptides), ORF7b (43aa, 3 peptides), ORF8 (121aa, 12 peptides), ORF10 (43aa, 3 peptides) proteins with a DMSO concentration of 0.2% (Fig. 1b). Experimental controls included: cytomegalovirus (CMV) peptide pool[16] and PMA/ionomycin as positive controls and for negative controls media alone and average DMSO control (1.0% concentration) for background cytokine production (Supplementary Figure 1b). SARS-CoV-2 peptide Megapools (Spike plus all pool, 467 peptides) for predicted MHC restricted peptides covering all proteins of the genome for CD4+ T cells and CMV from Grifoni et al. were used as initial positive controls[16].

**SARS-CoV-2-specific T cell intracellular cytokine staining**. Cryopreserved PBMCs were thawed and re-stimulated with overlapping peptide pools representing the SARS-CoV-2 structural proteins, accessory proteins, or ORF1ab (300 nM), DMSO (1% in RPMI), CMV peptide pool, PMA/ionomycin (1% in PBS) or RPMI alone for 6 h at 37 ℃. Golgi Plug (BD) containing Brefeldin A (1% in PBS), and Golgi Stop (BD) containing Monensin (0.67% in PBS) were added at 2 h during stimulation. Cells were stained (all antibodies from Biolegend, catalogue number, clone and dilution used) with Zombie-NIR (423106, 1:1,000) followed by anti-human CD3-PE/Dazzle 594 (980006, UCHT1, 1:200), CD4-BV605 (317438, OKT4, 1:100), CD8-AlexaFluor700 (344724, SK1, 1:100), CCR7-PerCP/Cy5.5 (353220, G043H7, 1:20), PD-1-BV421 (367422, NAT105, 1:50), CD25-PE (302606, BC96, 1:100) and CD45RA-APC (983004, HI100, 1:200) and a dump channel containing CD19-BV510 (302242, HIB19, 1:100), CD56-BV510 (318340, HCD56, 1:100) and CD14-BV510 (301842, M5E2, 1:100). Cells were then permeabilised and fixed (BD Cytofix/cytoperm) and further stained for anti-human IFNγ-FITC (502506, 4 S.B3, 1:50), IL-2-PECy7 (500326, MQ1-17H12, 1:50), TNF-BV711 (502940, MAb11, 1:50). For experiments including IL4 staining, an alternative

antibody cocktail was used, following cell permeabilisation, intracellular staining with anti-IFNγ-FITC, IL4-PE (500810, MP4-25D2, 1:50) and TNF-BV711 was carried out before acquisition of samples.

Activation-induced markers were detected following stimulation with the Structural peptide pool at a concentration of 2 μM or DMSO (1% in RPMI) with CD154-PE/Dazzle 594 (310840, 24-31, 1:100) for 18 h. Zombie-NIR staining was followed by anti-human CD4-BV605, CD8-AlexaFluor700, CCR7-PerCP/Cy5.5, PD-1-BV421, and CD45RA-APC, with the same dump channel were used again, with the addition of anti-human OX40-PE (350004, ACT35, 1:100), CD137-BV711 (309832, 4B4-1, 1:50), CD69-FITC (310904, FN50, 1:50) and CXCR5-PE-Cy7 (145516, L138D7, 1:50). Stained cells were acquired via flow cytometry (AttuneNxT) and analysed by FlowJo v10 (Supplementary Figure 1). IFNγ and AIM experiments were repeated twice on independent samples.

**Immunostaining of monocytes, T follicular helper cells, and plasmablasts**. Whole blood samples were stained with an antibody panel (all Biolegend and clone used) and live/dead Zombie-NIR to identify monocytes, Tfh, and plasmablast responses (Supplementary Figure 3). The combined monocytes/plasmablast panel contained: anti-human CD16-PE (980102, 3G8, 1:200), CD14-PerCPCy5.5 (301824, M5E2, 1:100), HLA-DR-BV605 (307640, L243, 1:75), CCR2-APC (357208, K035C2, 1:100), CD19-BV510 (302242, HIB19, 1:100), CD27-FITC (356404, M-T271, 1:50) and CD38-BV421 (303526, HIT2, 1:100). The Tfh panel contained: anti-human CD4-AlexaFluor700 (344622, SK3, 1:100), CXCR5-PerCPCy5.5 (356910, J252D4, 1:50), CD45RA-FITC (983002, HI100, 1:200), PD-1-BV711 (329928, EH12.2H7, 1:50) and ICOS-PE (313508, C398.4 A, 1:50). Cells were acquired by flow cytometry (AttuneNxT) and analysed by FlowJo v10 (Supplementary Figure 3).

**Spike-IgG quantification by ELISA**. Plates (Nunc MaxiSorp, Thermofisher Scientific) were coated with one representative coronavirus Spike protein at a time. Plates were coated with either 80 ng/ml of purified baculovirus-expressed Spike protein from 229E, NL63, HKU-1, and OC43 (SinoBiological). Plates were rinsed, blocked with 1% FBS in PBS, incubated with 1:100 HI plasma diluted in 0.05% Tween-20/ 0.1% FBS in PBS for 2 h then rinsed again, and incubated for 2 h with IgG-HRP (555788, G8-185, 1:5000; BD). HRP was revealed by stabilised hydrogen peroxide and tetramethylbenzidine (R&D systems) for 20 min, stopped with 2 N sulphuric acid and absorbance values were recorded at 450 nm on a spectrophotometer (Tecan Life Sciences).

**Statistics and reproducibility**. Statistical analysis was performed on Prism 9 (Graphpad). For two-way comparison, the Wilcoxon signed-rank test (paired) or Mann–Whitney *t*-test (unpaired) was used. For multiple-group comparisons, a Friedman (paired) or Kruskal–Wallis (unpaired) test, followed by the Dunn-Bonferroni post-hoc test was used. The One-sample Wilcoxon signed-rank test was used for comparisons against a hypothetical value of 1 for fold changes. Correlations were performed using the Spearman's test. To account for correlation due to multiple measurements from the same patients, a linear random-effects model was fitted (Supplementary Table 1). The model also tested the linear time trend by days after illness onset, and potential differences by age, sex, and symptomatic patients.

Differences were tested using Mann–Whitney test. Differences in baseline characteristics were detected with the chi-square test. Adjusted *p* values < 0.05 were considered statistically significant. Experiments were repeated successfully at least twice on independent samples.

**Reporting summary**. Further information on research design is available in the Nature Research Reporting Summary linked to this article.

## Data availability

The protein and peptide sequences and other data that support the findings of this study are available from the corresponding author upon reasonable request. The amino acid sequence of the peptide pools was based on βCoV/Hong Kong/VM20001061/2020 strain deposited in GenBank under accession code MT547814.1. Source data are provided in a Source Data file. Source data are provided with this paper.

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

## Acknowledgements

This study was partly supported by the Theme based Research Grants Scheme (T11-712/19-N), Health and Medical Research Fund (HMRF COVID-190115 and COVID-

190126), National Institutes of Allergy and Infectious Diseases, National Institutes of Health (USA) (HHSN272201400006C and U01AI151810). We thank Daniela Weiskopf, Jose Barrera, Shane Crotty, and Alessandro Sette from La Jolla Institute, USA, for providing SARS-CoV-2 peptide Megapools. This project utilised an Invitrogen Attune flow cytometer assisted by the Pasteur Foundation Asia.

## Author contributions

C. A. C. performed experiments. C. A. C., A. P. Y. L., A. H., N. K. and S. A. V. designed experiments. M. Y. W. K., W. H. C., Y. S. Y., S. S. C. O. T. Y. T., D. S. C. H. provided clinical information and supplied clinical samples, processed by F. N. L. M. coordinated by S. M. S. C. E. H. Y. L. performed further analysis. C. A. C., L. P., J. S. M. P. and S. A. V. designed the study. C. A. C., A. P. Y. L., A. H., N. K., L. P., J. S. M. P. and S. A. V. interpreted results and wrote the paper.

## Competing interests

The authors declare no competing interests.
