## [Peer Review File · Nature Communications]

REVIEWER COMMENTS

Reviewer #1 (Remarks to the Author):

The study by Cohen et al quantified SARS-CoV-2-specific immune responses in children and adults. The authors found that while CD4+ T cell responses increased with age, CD8+ T cells increased with time. Interestingly, children had lower CD4+ and CD8+ T cell responses when compared to adults. T cell specificities were also skewed differentially towards peptides derived from different SARS-CoV-2 compartments. Children had lower level of SARS-CoV-2-specific antibodies and monocytes in blood, however follicular T helper cells were increased. The authors conclude that reduced prior β -coronavirus immunity and activation of de novo immune responses in children might potentially result in milder COVID-19 pathogenesis. This is a very important, timely and well written study performing comprehensive immune analysis to depict differences in immune responses between COVID-19 children and adults.

Specific comments:

1. Fig 1: The authors are encouraged to include representative FACS plots for both children and adults.
2. It seems that the interesting data on maximum increases of IFN+ CD4+ and CD8+ T cells with age (Supp Fig 2) and cellular recruitment during acute disease (Supp Fig 4) are 'hidden' as supplementary data. The authors should consider including these findings as main figures.
3. Fig 3AB: would the authors have data for negative children? I do understand that the pediatric PBMCs are more difficult to obtain than adult PBMCs.
4. Discussion seems a bit lengthy and should be shorten.

Reviewer #2 (Remarks to the Author):

In this study, the authors assess the SARS-CoV-2-specific T cell responses and other immune cell parameters in blood samples from a cohort of children and adults with mild COVID-19, during the acute phase and at different timepoints after recovery. They also measure some other parameters, including monocyte frequencies and subsets, T-follicular-helper (Tfh) frequencies, and the presence of antibodies-specific for seasonal coronaviruses. They find a reduced CD4+ and CD8+ T cells response (in terms of frequencies of IFN- γ + cells following peptide stimulation) specific for SARS-CoV-2 structural proteins in children compared to adults, while T cell responses specific for accessory and non-structural proteins did not differ. The overall functional responses of virus-specific T cells was not different between children and adults. There were increased TFh cells in children compared to adults, and some differences in overall monocyte frequencies. Overall, the experiments seem well-performed and thorough in the assessment of the different cohorts over time. However, there certain of the differences identified between children and adults are difficult to interpret, because of the lack of a pediatric control population (The control population is all adults). In addition, the readout of IFN- γ for the T cell assays is likely to underestimate the virus-specific CD4 T cell response. Specific comments are listed below.

1. There are certain inherent differences in T cells between children and adults, including the increased frequency of naive T cells, and reduced frequency of effector-memory (TEM) cells, that are important to consider in their experimental setup and therefore require a pediatric, non-SARS-CoV-2 infected control population. In particular, the measurements of SARS-CoV-2-specific CD8 T cells responses showed no increase in the response of children or adults, compared to adult controls; however, the authors still scored, plotted and analyzed these results as virus-specific responses in Figure 1I, J, K, Figure 2B, D, G, J, and other supplementary figures. Since these were no different than non-infected individuals (At least for adults), how can these be scored as SARS-CoV-2 specific? It would be important to measure the CD8 T cells responses to SARS-CoV-2 peptides in children who were not infected, as they would be less likely to have a cross-reactive memory CD8 T cell response.

Other important aspects of their study that require the non-infected pediatric controls to interpret, are the differences in overall IFN-gamma production from PMA stimulation between children and adults shown in Supplementary figure 2-- that likely derives from the increased frequency of memory T cells in adult compared to children's blood. The differences in monocyte frequencies between children and adults in supplementary figure 4 is difficult to interpret without the pediatric control, as well as the total Tfh frequencies. When comparing pediatric versus adult immune responses, it is important to have control groups for each, due to inherent differences between the two groups that are still not fully understood.

2. The information in supplementary Figure 4 could be part of the main figures, particularly the Tfh frequency analysis.

3. The T cell assays used IFN-gamma as the readout. Many of these assays also use the activation-induced marker (AIM) assay of Crotty and colleagues. This assay allows for a more complete measurement of virus-reactive T cells without biasing for specific functional readouts, which could underestimate their frequencies. Also, memory CD4 T cells do not always produce IFN-gamma, so this readout could be underestimating virus-specific CD4 T cell frequencies. Did the authors quantitate T cell responses using activation markers as a readout? Were the results similar or were the frequencies for certain epitopes increased?

4. The rationale for measuring the antibody response to other types of corona viruses is not clear, and the results do not provide insight into differences between pediatric and adult immune responses. This figure (Figure 4) could be put in supplement.

Reviewer #1 (Remarks to the Author):

The study by Cohen et al quantified SARS-CoV-2-specific immune responses in children and adults. The authors found that while CD4+ T cell responses increased with age, CD8+ T cells increased with time. Interestingly, children had lower CD4+ and CD8+ T cell responses when compared to adults. T cell specificities were also skewed differentially towards peptides derived from different SARS-COV-2 compartments. Children had lower level of SARS-CoV-2-specific antibodies and monocytes in blood, however follicular T helper cells were increased. The authors conclude that reduced prior β -coronavirus immunity and activation of de novo immune responses in children might potentially result in milder COVID-19 pathogenesis. This is a very important, timely and well written study performing comprehensive immune analysis to depict differences in immune responses between COVID-19 children and adults.

Specific comments:

1. Fig 1: The authors are encouraged to include representative FACS plots for both children and adults.

We have now added representative FACS plots to Fig 1 of children's IFN γ T cell responses in red.

2. It seems that the interesting data on maximum increases of IFN+ CD4+ and CD8+ T cells with age (Supp Fig 2) and cellular recruitment during acute disease (Supp Fig 4) are 'hidden' as supplementary data. The authors should consider including these findings as main figures.

We thank the reviewer for the emphasis on this important results and have now moved these supplementary figures to the main text.

We have now moved maximum increases of IFN+ CD4+ and CD8+ T cells with age (original Supp Fig 2) to Figure 3.

We have now moved cellular recruitment during acute disease (original Supp Fig 4) to Figure 4.

3. Fig 3AB: would the authors have data for negative children? I do understand that the pediatric PBMCs are more difficult to obtain than adult PBMCs.

In anticipation of this comment we had begun recruiting uninfected children since January 2021. Due to limited samples, and the fact we had already used the majority of infected children's samples at acute timepoints, we have made limited but useful comparisons with uninfected children and infected children for specific (Figure 1G), background and maximal T cell activation (Figure 3) and cellular recruitment (Figure 4G-J).

4. Discussion seems a bit lengthy and should be shorten.

The discussion has been trimmed throughout for repetition to reduce length (original discussion: 1,570 words -> revised: 1,103 words). Please see track change version for text deletions.

Reviewer #2 (Remarks to the Author):

In this study, the authors assess the SARS-CoV-2-specific T cell responses and other immune cell parameters in blood samples from a cohort of children and adults with mild COVID-19, during the acute phase and at different timepoints after recovery. They also measure some other parameters, including monocyte frequencies and subsets, T-follicular-helper (Tfh) frequencies, and the presence of antibodies-specific for seasonal coronaviruses. They find a reduced CD4+ and

CD8+T cells response (in terms of frequencies of IFN-gamma+ cells following peptide stimulation) specific for SARS-CoV-2 structural proteins in children compared to adults, while T cell responses specific for accessory and non-structural proteins did not differ. The overall functional responses of virus-specific T cells was not different between children and adults. There were increased TFh cells in children compared to adults, and some differences in overall monocyte frequencies. Overall, the experiments seem well-performed and thorough in the assessment of the different cohorts over time.

However, there certain of the differences identified between children and adults are difficult to interpret, because of the lack of a pediatric control population (The control population is all adults). In addition, the readout of IFN-gamma for the T cell assays is likely to underestimate the virus-specific CD4 T cell response. Specific comments are listed below.

To address this comment, and those by Reviewer 1 also, we have added IL-4 and AIM assay measures (Figure 1G), and an uninfected children control group. Across 3 antigen specific assays (IFN γ , IL-4 and AIM), and normalisation by maximal activation, infected children had lower magnitude T cell responses than infected adults.

1. There are certain inherent differences in T cells between children and adults, including the increased frequency of naive T cells, and reduced frequency of effector-memory (TEM) cells, that are important to consider in their experimental setup and therefore require a pediatric, non-SARS-CoV-2 infected control population. In particular, the measurements of SARS-CoV-2-specific CD8 T cells responses showed no increase in the response of children or adults, compared to adult controls; however, the authors still scored, plotted and analyzed these results as virus-specific responses in Figure 1I, J, K, Figure 2B, D, G, J, and other supplementary figures. Since these were no different than non-infected individuals (At least for adults), how can these be scored as SARS-CoV-2 specific? It would be important to measure the CD8 T cells responses to SARS-CoV-2 peptides in children who were not infected, as they would be less likely to have a cross-reactive memory CD8 T cell response.

We would like to clarify the data for this reviewer comment, as SARS-CoV-2 T cells are found in healthy donors due to cross-reactivity and increase in magnitude after infection in a number of studies by others already. In Figure 1F, we saw a significant fold change increase in structural specific CD8 T cells between hospital admission and discharge for paired adult samples ($p=0.023$, fold change: 70.64 +/- 214.1 mean +/- SD, Median: 1.724). This was not significant however in the comparison of the average response due to heterogeneity (Figure 1E), and was less pronounced than the increase in CD4 T cell responses (Figure 1DF) ($p= 0.0005$, fold change 10.79 +/- 17.07 mean +/- SD, Median: 3.237). Furthermore, the magnitude of the acute CD8 T cell response was not different between adults and negative control adults (Figure 2B), however over time structural specific CD8 T cells increased (Figure 5B), leading to significant differences at memory (day >14 post infection) (Figure 5H). The majority of healthy unexposed adults have pre-existing cross-reactive CD8 T cells for SARS-CoV-2, which was also observed by others (studies by La Bert, Mateus, Peng, Grifoni). Therefore, SARS-CoV-2 specific CD8 T cell responses are real and detected by our assays but are not substantially recruited by SARS-CoV-2 infection. This likely plays an important role in the long term pathogenesis of COVID-19 and shedding of virus (Discussion page 15, line 538-548).

Uninfected children negative controls are now included in our study. However due to limited samples, and the fact we had already used the majority of infected children's samples at acute timepoints, we have made limited but useful comparisons with uninfected children and infected children for specific (Figure 1G), background and maximal T cell

activation (Figure 3) and cellular recruitment (Figure 4G-J). Children also had lower CD4⁺ T cell background, but higher maximal CD4⁺ and CD8⁺ T cell activation than uninfected adults, indicating inherent differences in activation thresholds. Direct comparison of infected children (at memory timepoints) did not have significantly increased CD4⁺ or CD8⁺ T cell responses compared to uninfected children (Figure 1G), but further normalisation by maximal PMA/ionomycin responses shows infected children have increased CD4⁺ T cell responses compared to uninfected children (Figure 3E), however even with normalisation (of maximal responses and background DMSO subtraction) infected children still have lower CD4⁺ and CD8⁺ T cell responses than infected adults. Furthermore, uninfected children had significantly lower IFN γ ⁺ CD8⁺ T cell responses than uninfected adults (Figure 1G), which is in agreement with this reviewer and our presumption that children have lower cross reactive CD8 T cell memory with lower common cold coronavirus exposure.

Other important aspects of their study that require the non-infected pediatric controls to interpret, are the differences in overall IFN-gamma production from PMA stimulation between children and adults shown in Supplementary figure 2-- that likely derives from the increased frequency of memory T cells in adult compared to children's blood. The differences in monocyte frequencies between children and adults in supplementary figure 4 is difficult to interpret without the pediatric control, as well as the total Tfh frequencies. When comparing pediatric versus adult immune responses, it is important to have control groups for each, due to inherent differences between the two groups that are still not fully understood.

Uninfected children PBMC have now been added to our study and comparisons made with infected children at memory timepoints (as discussed above). We found that uninfected adults and children had comparable monocytes, but increased Tfh and plasmablast responses in uninfected children, therefore there is an increased adaptive response recruitment potential in children.

Due to the inherent thresholds of activation between adults and children we further normalised T cell data after subtraction of DMSO background to % of maximum activation by PMA stimulation (Figure 4). Infected children still had significantly higher CD4⁺ T cell responses than uninfected children, but lower responses than infected adults.

2. The information in supplementary Figure 4 could be part of the main figures, particularly the Tfh frequency analysis.

We thank the reviewer for the emphasis on this important results and have now moved this supplementary figures to the main text.

We have now moved cellular recruitment during acute disease (original Supp Fig 4) to Figure 4.

3. The T cell assays used IFN-gamma as the readout. Many of these assays also use the activation-induced marker (AIM) assay of Crotty and colleagues. This assay allows for a more complete measurement of virus-reactive T cells without biasing for specific functional readouts, which could underestimate their frequencies. Also, memory CD4 T cells do not always produce IFN-gamma, so this readout could be underestimating virus-specific CD4 T cell frequencies. Did the authors quantitate T cell responses using activation markers as a readout? Were the results similar or were the frequencies for certain epitopes increased?

We thank the reviewer for the alternate marker suggestion, and have now performed these additional assays. Following structural pool peptide stimulation, IFN- γ (Th1) IL-4 (Th2) and activation induced markers (AIM by CD40L⁺ CD69⁺ CD137⁺ OX40⁺) for T cell responses were assessed. Across 3 antigen specific assays (IFN γ , IL-4 and AIM), and normalisation by maximal activation, infected children had lower magnitude T cell responses than infected adults.

This is now included in the results section (page 7 and 9) discussion (page 13).

Results, page 7:

“To confirm the appropriate use of IFN γ production as a surrogate measure of virus specific T cell responses, three assays were initially used. T cell responses from infected children and adults (memory timepoint samples, > day 14) and negative controls of both children and adults’ for CD4⁺ T cells were measured for IFN γ production, IL-4 production, and expression of activation induced markers (AIM by CD40L⁺ CD69⁺ CD137⁺ OX40⁺, negative adults only) and CD8⁺ T cells responses were measured by IFN γ production and AIM expression (Figure 1G). IFN γ and AIM assays also showed higher responses in infected adults compared with negative adults confirming assay specificity. All assays showed that infected adults had greater structural specific T cell responses than infected children. However, the low magnitude of T cell responses directly *ex vivo* from infected children showed no significant difference to uninfected children. IFN γ CD8⁺ T cells responses further showed a difference between negative children and negative adults, but there was no difference between infected and control groups of either age group. Uninfected children had significantly lower IFN γ ⁺ CD8⁺ T cell responses than uninfected adults (Figure 1G), which is in agreement with our hypothesis that children have lower cross reactive CD8⁺ T cell memory with lower prior common cold coronavirus exposure. IFN γ and IL-4 T cell responses are distinct cell populations with no correlation between responses (Figure 1H).”

Results, page 9:

“However, baseline differences exist between adults and children for non-specific T cell activation³⁰⁻³². The baseline activation (by DMSO) (Figure 3A) and overall maximum activation (by PMA/ionomycin) (Figure 3B) is lower in infected children. Overall background and maximum T cell responsiveness significantly increases with age in infected subjects (Figure 3CD). Adult negative controls had comparable background IFN γ induction compared to infected adults (Figure 3A), but significantly higher maximum responsiveness (Figure 3B). Whereas negative children had significantly higher non-specific (for CD4⁺ T cells) and maximum activation than infected children, yet Structural specific IFN γ ⁺ T cell activation was no different between these groups (Figure 1G). Therefore, normalisation of structural specific T cells by % of maximum PMA/ionomycin responses after background DMSO subtraction shows that infected adults continue to have higher virus specific CD4⁺ and CD8⁺ IFN γ T cell responses than children, whilst negative adults also have a higher responses than negative children. This normalised response shows a significant difference between infected and uninfected children, and infected and uninfected adults for the CD4⁺ but not CD8⁺ IFN γ T cell response (Figure 3E). Overall, even with normalisation infected children still have lower CD4⁺ and CD8⁺ T cell responses than infected adults. The fine specificity of identifying low frequency antigen specific T cells directly *ex vivo* may be obscured through normalisation, and maximal activation is refractory to recent infection, therefore T cell responses should be considered directly *ex vivo* with paired DMSO background subtracted.”

4. The rationale for measuring the antibody response to other types of corona viruses is not clear, and the results do not provide insight into differences between pediatric and adult immune responses. This figure (Figure 4) could be put in supplement.

We thank the reviewer for this constructive comment, however, would prefer to keep this figure as a main figure. The rationale of this data is to address the importance of prior immunity of infected children versus infected adults, given the differences in T cell response magnitude and potential for T cell cross-reactivity.

Our T cell data indicated a difference in T cell memory between children and adults (also described in reviewer 2 comment 1), and a growing body of evidence (studies by La Bert, Mateus, Peng, Grifoni) suggests SARS-CoV-2 T cell cross reactivity in healthy adults (Figure 1G). Pre-existing T cell cross reactivity is potentially generated through prior exposure to related coronaviruses, we found no difference in α -coronavirus specific antibodies, but β -coronavirus antibodies were significantly lower in children than adults. However, the correlation of β -coronavirus prior immunity by antibodies and SARS-CoV-2 specific CD4 and CD8 T cell responses was not significant. The prior immunity data supported by serology experiments is important as it supports our data that children have lower T cell response magnitude due to lower baseline responses rather than a defect in T cell recruitment or on a per cell basis (based on cytokine quality and fold changes in responses).

REVIEWERS' COMMENTS

Reviewer #1 (Remarks to the Author):

The authors have addressed all my comments. This manuscript is an outstanding addition to the COVID-19 literature.

Reviewer #2 (Remarks to the Author):

The authors have addressed the comments by adding analysis of virus-reactive T cells and overall functional analysis of T cells in an uninfected pediatric cohort, and other controls. They find a significant background for assessing SARS-CoV-2-specific T cells in uninfected subjects, but significant differences between children and adults. The high background is more than what has been observed in other studies. They also find differences in the overall functional capacity of children and adults, although interestingly, the uninfected children have higher IFN-gamma responses to stimulation with the non-specific stimuli, PMA/ionomycin, compared to uninfected children? Was this due to a certain subsets--i.e. which T cells subset was this higher proportion of IFN-gamma produced by? This is an interesting finding.

Overall, the results show a detailed comparison of immune parameters between children and adults to SARS-CoV-2 infection.

REVIEWERS' COMMENTS

Reviewer #1 (Remarks to the Author):

The authors have addressed all my comments. This manuscript is an outstanding addition to the COVID-19 literature.

We thank the reviewer for this positive reply.

Reviewer #2 (Remarks to the Author):

The authors have addressed the comments by adding analysis of virus-reactive T cells and overall functional analysis of T cells in an uninfected pediatric cohort, and other controls. They find a significant background for assessing SARS-CoV-2-specific T cells in uninfected subjects, but significant differences between children and adults. The high background is more than what has been observed in other studies.

The DMSO background was subtracted and negative unexposed control included throughout. The high background compared to other studies may be due to differing cohorts and assays used, hence direct comparison of data values is difficult between different studies.

They also find differences in the overall functional capacity of children and adults, although interestingly, the uninfected children have higher IFN-gamma responses to stimulation with the non-specific stimuli, PMA/ionomycin, compared to uninfected children? Was this due to a certain subsets--i.e. which T cells subset was this higher proportion of IFN-gamma produced by? This is an interesting finding.

The higher PMA responses in uninfected children compared to infected controls may be due to residual inflammation following infection, which has been noted in other studies that PMA stimulation is refractory to HIV infection (Crawford et al., 2013) and severe H1N1 pandemic infection (Agarti et al., 2010). It is difficult to pinpoint the attributing factor without comparison to the after effects of other stimuli (vaccination) or infections. The PMA/ionomycin IFN γ ⁺ CD4⁺ T cell subsets were not significantly different between cohorts and were derived mostly from effector memory T cells.

Overall, the results show a detailed comparison of immune parameters between children and adults to SARS-CoV-2 infection.

We thank the reviewer for this positive reply.

Editorial comments are addressed in the extended reply. Changes to the figure lettering (ABC -> abc), figure and section headings, abstract length, statistical test updates, catalogue and dilution details for antibodies, parental consent, age and gender of uninfected children controls, and updated reporting summary.

The entire source data for each figure panel is not available at this time of resubmission and in the data availability statement it is available from the corresponding author upon request. If this impedes the publication of this manuscript we can prepare the source data file but need further time.

Reporting summary

Q1. Statistics: Please see the extended comments document for a list of figures and figure legends that require additional information.

The p values and stat test have been updated in the legends and text.

Q2. Please consider making all the protein, peptide sequences and the data that support the findings of this study, available in a publicly accessible repository, or explain to the editor why the data can only be made available from the authors on request.

Please provide a valid and accessible identifier to the data deposited in GISAID with ID: EPI_ISL_412028.

The protein sequence is already publicly available, the GenBank code and website for the peptides has been updated in the Data availability section.

From:

The protein, peptide sequences and data that support the findings of this study are available from the corresponding author upon request. The amino acid sequence of the peptide pools was based on β CoV/Hong Kong/VM20001061/2020 strain (GISAID ID: EPI_ISL_412028). Data from flow cytometry and ELISA IgG responses with background subtracted are indicated, and representative flow cytometry plots are shown.

Updated to:

The protein, peptide sequences and data that support the findings of this study are available from the corresponding author upon reasonable request. The amino acid sequence of the peptide pools was based on β CoV/Hong Kong/VM20001061/2020 strain under accession code GenBank: MT547814.1 [<https://www.ncbi.nlm.nih.gov/nuccore/MT547814>]. Data from flow cytometry and ELISA IgG responses with background subtracted are indicated in all figures, and representative flow cytometry plots are shown.

Q3. Replication: Please confirm if all attempts at replication were successful.

Experiments were repeated at least twice on independent samples. Updated in methods statistics and reproducibility section.

Q4. Antibodies used: Please provide the catalog number for all the antibodies used in the study. Also, please state the dilutions of the antibodies in the manuscript. For example: anti-human CD3-PE/Dazzle 594, etc.

Updated in reporting summary and in the methods

Q5. Validation: Please elaborate on the validation of all primary antibodies for the application, noting any validation statements on the manufacturer's website. Please include this information directly in the reporting summary.

From: Product specification sheets for each antibody describe the expected cell proportion. All antibodies were titrated prior to use on

PBMC samples from uninfected healthy donors, and activation markers assessed after PMA/ionomycin stimulation. Single colour fluorescence controls are acquired for each antibody at the time of data acquisition.

Updated to: Product specification sheets for each antibody describe from the company the expected cell proportion based on cell staining. All antibodies were titrated by us prior to use on patient PBMC samples from uninfected healthy donors to determine optimal staining concentrations, and activation markers assessed after PMA/ionomycin stimulation. Single colour fluorescence controls are acquired for each antibody at the time of data acquisition to ensure antibodies in use are working well.

Q6. Population characteristics: Please describe the covariate-relevant population characteristics of the human research participants (e.g. age, gender).

From: We used opportunistic sampling of RT-PCR COVID-19 patients that were available in Hong Kong were recruited for this study. No sample size calculation was performed. An initial sample size of n=24 children, 45 adults SARS-CoV-2 patients and n=15 children and 25 adults age- and sex-matched controls was used in our study.

Updated to:

RT-PCR-confirmed infected patients were recruited to participate in immune studies of COVID-19. All patients and the parents of children provided informed consent. RT-PCR+ COVID-19 patients in Hong Kong were recruited for this study through their consulting physician. An initial sample size of infected n=24 children (mean stdev: 8.1±3.9, range: 1.92 (23 months)-13 years), 45 adults (mean±stdev: 43.1±13.7, range: 20-65 years) SARS-CoV-2 patients and uninfected n=15 children (10.3±3.2, 2-14 years) and 25 adults (37.6±13.0, 19-57 years) age- and sex-matched controls were used in our study. The details of their age, gender and patient symptom severity is given in Table 1.

Q7. Recruitment: Please elaborate on how the participants were recruited.

From:

RT-PCR confirmed COVID-19 infection were enrolled during clinical care in hospitals in Hong Kong (China, SAR) and all of them provided informed consent. Their blood was collected at various time-point after disease onset, and there was no bias to the recruitment or collection.

Updated to:

RT-PCR confirmed COVID-19 infection were enrolled during clinical care in hospitals in Hong Kong (China, SAR) and all of them provided informed consent. Their blood was collected at various time-point after disease onset, including hospital admission and discharge and long term follow up with clinicians, and there was no bias to the recruitment or collection. Samples were selected from a large biobank of patients for time point similarity between infected adults and children, and age matching for negative controls.

Q8. Ethics oversight: Please provide a complete statement on ethical approval here in the reporting summary as provided in the manuscript.

Please provide a statement on informed consent from parents/LAR of SARS-CoV-2 seropositive children recruited in the study.

From:

The COVID-19 patient study was approved by the institutional review board of the Hong Kong West Cluster of the Hospital Authority of Hong Kong (approval number: UW20-169), and all of patients provided informed consent.

This negative control samples was from Hong Kong blood donors was approved by the Institutional Review Board of The Hong Kong University and the Hong Kong Island West Cluster of Hospitals (IRB reference number UW16-254), and all of participants provided informed consent.

Updated to:

The study was approved by the institutional review board of the respective hospitals, viz. Kowloon West Cluster (KW/EX-20-039 (144-27)), Kowloon Central/Kowloon East cluster (KC/KE-20-0154/ER2) and HKU/HA Hong Kong West Cluster (UW 20-273, UW20-169), Joint Chinese University of Hong Kong-New Territories East Cluster Clinical Research Ethics Committee (CREC 2020.229). All of patients, and children and their parents provided informed consent. The collection of SARS-CoV-2 seronegative adult negative control blood donors (37.6 ± 13.0 , 19-57 years) was approved by the Institutional Review Board of The Hong Kong University and the Hong Kong Island West Cluster of Hospitals (UW16-254). SARS-CoV-2 seronegative children's control blood donors ($n=15$) were recruited from immunocompetent children from renal, endocrine and blood clinics (10.3 ± 3.2 , 2-14 years) who were donating blood for non-infection related purposes. Informed consent was given by patients and parents and the collection of these samples was approved by HKU/HA Hong Kong West Cluster Hospitals (UW 20-273, UW20-169).